# A Novel Chemotherapy Combination to Enhance Proteotoxic Cell Death in Hepatocellular Carcinoma Experimental Models Without Killing Non-Cancer Cells

**DOI:** 10.3390/ijms26146699

**Published:** 2025-07-12

**Authors:** Carlos Perez-Stable, Alicia de las Pozas, Teresita Reiner, Jose Gomez, Manojavan Nagarajan, Robert T. Foster, Daren R. Ure, Medhi Wangpaichitr

**Affiliations:** 1Research Service, Bruce W. Carter Veterans Affairs Medical Center, Miami, FL 33125, USA; delaspozasalicia@gmail.com (A.d.l.P.); teresitare@msn.com (T.R.); jose.gomez1@va.gov (J.G.); mxn816@miami.edu (M.N.); mwangpaichitr@med.miami.edu (M.W.); 2South Florida VA Foundation for Research and Education, Miami, FL 33125, USA; 3Geriatric Research, Education, and Clinical Center, Bruce W. Carter Veterans Affairs Medical Center, Miami, FL 33125, USA; 4Department of Medicine, Division of Gerontology & Palliative Medicine, University of Miami Miller School of Medicine, Miami, FL 33136, USA; 5Sylvester Comprehensive Cancer Center, University of Miami Miller School of Medicine, Miami, FL 33136, USA; 6Hepion Pharmaceuticals, Edmonton, AB T5J 4P6, Canadadaren.ure@gmail.com (D.R.U.); 7Department of Surgery, Division of Cardiothoracic Surgery, University of Miami Miller School of Medicine, Miami, FL 33136, USA

**Keywords:** hepatocellular carcinoma, proteotoxic stress, apoptosis, cyclophilin, proteasome, endoplasmic reticulum stress, unfolded protein response

## Abstract

Inhibitors of the ubiquitin–proteasome system increase proteotoxic stress and have achieved clinical success for multiple myeloma but not for solid cancers such as hepatocellular carcinoma. Our objective is to identify a combination with proteasome inhibitors that enhances proteotoxic stress and apoptotic cell death in hepatocellular carcinoma but with less toxicity to non-cancer cells. We found that rencofilstat, a pan-cyclophilin inhibitor, combined with ixazomib, a proteasome inhibitor, increased apoptotic cell death in hepatocellular carcinoma but not in umbilical vein or dermal fibroblast non-cancer cells. We then analyzed the effects of rencofilstat + ixazomib on XBP1s and PERK, critical factors in the unfolded protein response used by cells to survive proteotoxic stress. Rencofilstat + ixazomib maintained higher expression of XBP1s and genetic models suggested that XBP1s was a pro-survival protein early and pro-death protein at later times. Simultaneously, decreased PERK expression prevented the block in protein synthesis via phospho-eIF2α and likely further amplified proteotoxic stress. Rencofilstat + ixazomib did not have effects on XBP1s or PERK in non-cancer cells. Further genetic experiments revealed the pro-survival roles for cyclophilin A and B in mediating rencofilstat + ixazomib-induced cell death. In the Hep3B xenograft model, rencofilstat + ixazomib significantly inhibited tumor volumes/weights without general toxicity. We conclude that rencofilstat + ixazomib amplified proteotoxic stress in hepatocellular carcinoma past a threshold pro-survival pathways could not tolerate, whereas non-cancer cells were less affected.

## 1. Introduction

A “hallmark of cancer” is the ability to overcome the harmful effects of proteotoxic stress resulting from enhanced protein synthesis required for tumor growth [1,2,3]. Inhibitors of the ubiquitin–proteasome system (UPS, the major pathway that degrades unfolded proteins [4]) increase proteotoxic stress and have achieved clinical success for multiple myeloma but not for solid cancers such as hepatocellular carcinoma (HCC) [5,6,7]. In addition, the dose-limiting toxic side effects of UPS inhibitors complicate their clinical use. Unlike in highly secretory multiple myeloma (chronic proteotoxic stress), it is expected that advanced solid cancers (less secretory) such as HCC require combinations with UPS inhibitors to further enhance proteotoxic stress past a tolerable threshold, resulting in apoptotic cell death. Since the increase in hepatitis C, obesity, and fatty liver disease has elevated HCC as the cancer with the fastest growing death rate in the US [8,9], the discovery of new combinations with UPS inhibitors to enhance proteotoxic stress and apoptotic cell death in HCC with less toxicity is an important area for research.

Unlike normal cells, cancer cells require increased endoplasmic reticulum (ER) activity to maintain protein homeostasis. ER stress is an imbalance between excessive unfolded/misfolded proteins in the ER and the capacity to refold these proteins. The unfolded protein response (UPR) is an adaptive pro-survival mechanism responding to ER stress that activates kinase signaling pathways (IRE1α, PERK, and ATF6) to stop further protein synthesis (increase phospho (P)-eIF2α) and enhance the folding capacity (increase XBP1s/CHOP/ATF4 positive transcription factors for chaperones) [10,11]. Because the UPR is already highly activated in cancer cells, further increasing ER stress may have an irreversible lethal effect on cancer cells but not on non-cancer cells. Since blocking the UPS leads to an accumulation of poly-ubiquitinated (poly-Ub) proteins, important roles for ER stress and the UPR are implicated in mediating apoptotic cell death induced by UPS inhibitors [12,13,14]. UPS inhibitors also trigger autophagy, an adaptive pro-survival homeostatic system that helps clear poly-Ub proteins and damaged organelles [15,16]. If the UPR and autophagy fail to protect from excessive proteotoxic stress, apoptotic cell death is triggered [10,11,15,16,17,18]. Although enhancing proteotoxic stress in solid cancers by combining heat shock protein 90 or pan-histone deacetylase + UPS inhibitors are promising strategies, clinical results indicate some responses but are accompanied by high toxicities [19,20].

We propose a novel strategy to enhance proteotoxic stress in HCC but with less toxic side effects on non-cancer cells by combining the pan-cyclophilin inhibitor rencofilstat (RCF, non-immunosuppressive analog of cyclosporin A, formerly known as CRV431) [21,22] and a second-generation UPS inhibitor (ixazomib, Ixz; approved for multiple myeloma) [23,24]. Cyclophilins (Cyps), a family of highly abundant proteins important in protein folding and stress response, are localized in multiple cellular compartments (cytosol, ER, mitochondria, and nucleus) and in extracellular secretions [25,26,27]. There is a clear rationale for using Cyp inhibitors against CypA and B isoforms in cancer chemotherapy because of their roles in tumor progression and stress protection [28,29]. Therefore, combining the Cyp inhibitor RCF to increase misfolded proteins and blocking their degradation with the UPS inhibitor Ixz should amplify proteotoxic stress (increased levels of poly-Ub proteins) and force apoptotic cell death (increased cleaved PARP levels). A significant finding in patients with highly resistant multiple myeloma supports this strategy [30].

In this report, we confirmed that the combination of RCF + Ixz had a synergistic anti-HCC effect without toxicity to non-cancer cells. Our results showed that RCF + Ixz increased apoptotic cell death, maintained high levels of XBP1s (UPR transcription factor; [10,11]), and decreased CypB levels. Inducible knockdown and expression experiments confirmed the pro-survival roles of CypA and CypB, whereas a more complex role for XBP1s (pro-survival early and pro-death late) was revealed. RCF + Ixz also blocked the ability of HCC cells to decrease protein synthesis during proteotoxic stress via the PERK/eIF2α pathway, suggesting that the maintenance of protein synthesis during acute ER stress is important for cell death. An important role for autophagy in increasing RCF + Ixz-induced cell death was also revealed by our data. Finally, we showed that orally bioavailable RCF + Ixz had an anti-HCC effect on a mouse model without toxicity. Overall, RCF + Ixz amplified ER and proteotoxic stress over a critical threshold to promote apoptotic cell death in HCC cells, while normal cells were less affected.

## 2. Results

### 2.1. The RCF + Ixz Combination Enhances Apoptotic Cell Death in HCC Cells but Has Less of an Effect on Non-Cancer Cells

We determined in the human HCC cell lines Hep3B, PLC/PRF/5, and HepG2 the extent of cell death after treatment with RCF + Ixz. The results indicated RCF + Ixz increased cell death in all HCC cells compared to RCF and Ixz alone (Figure 1A and Appendix A). The pan-caspase apoptosis inhibitor QVD reduced the RCF + Ixz-induced increases in cell death and cleaved (cl) PARP levels (a marker of apoptosis) in HCC cells, suggesting apoptosis (Figure 1A–C and Appendix A). Similarly, cyclosporin A (CsA) + Ixz also greatly increased cell death in Hep3B and HepG2 cells, suggesting that Cyp was important for this effect (Appendix A). A Calcusyn analysis of cell proliferation assays revealed that RCF or CsA + Ixz synergistically inhibited HCC cell proliferation (CI < 0.5 exceeding the synergism threshold of a CI < 0.7) (Appendix A). To determine if the same concentrations of RCF + Ixz that were lethal in HCC cells had effects on non-cancer cells, we obtained human EA.hy926 (EA, umbilical vein) cells and primary human dermal fibroblasts (HDFs). The results indicated RCF + Ixz had much less of an effect on cell death in EA cells (3.6%) and HDFs (1.9%) compared to PLC (78%) and Hep3B (68%) HCC cells (Figure 1D and Appendix A).

### 2.2. RCF + Ixz Increases Proteotoxic Stress and Alters Autophagy in HCC Cells

Western blot analyses (24 and 48 h) indicated that RCF + Ixz increased proteotoxic stress (poly-Ub), the UPR (XBP1s and IRE1α), and autophagy (LC3B) in HCC cells. Variable effects of RCF + Ixz on p62 levels (a marker of autophagy; [31]) suggested increased autophagy in Hep3B and PLC cells (p62 levels were decreased) and decreased autophagy in HepG2 cells (p62 levels were increased) (Figure 1B,C and Appendix A). RCF alone and RCF + Ixz decreased CypB levels but not CypA levels. Unlike in HCC cells, RCF + Ixz had little or no effect on cl-PARP, XBP1s, and LC3B levels in EA and HDF non-cancer cells (Figure 1E and Appendix A). However, similar to HCC cells, RCF + Ixz increased poly-Ub levels, suggesting that non-cancer cells may be more resistant to proteotoxic stress caused by higher poly-Ub levels. RCF + Ixz (and RCF alone) also decreased CypB levels, and there was no effect on CypA levels.

Analyses at earlier times in Hep3B and PLC cells revealed that RCF alone and RCF + Ixz increased XBP1s levels (0.5–2 h) and decreased CypB levels (4 h) whereas RCF + Ixz increased poly-Ub levels (2–4 h) and increased cl-PARP levels at later times (8 h) (Appendix A). In HepG2 and Hep3B cells, RCF alone produced immediate early increases in IRE1α (10 min) and XBP1s (1 h) levels followed by decrease in CypB levels (4 h) (Appendix A). The Q-PCR analysis of CypB suggested that the RCF + Ixz-induced decrease in protein levels was not due to decreased mRNA expression, whereas there was a small increase in XBP1s expression (Appendix A). Since CypB does not have an ER retention sequence, Cyp inhibitors often cause its extracellular secretion from cells, resulting in decreased intracellular levels [32,33]. These results suggested that RCF + Ixz increased apoptotic cell death in a variety of HCC cells, which correlated with the enhancement of proteotoxic stress, the UPR, and changes in autophagy.

### 2.3. Maintenance of Protein Synthesis by Blocking PERK/P-eIF2α Increases RCF + Ixz-Induced Proteotoxic Cell Death in HCC Cells

Cycloheximide (Chx), a protein synthesis inhibitor, blocked RCF + Ixz-induced cell death in Hep3B and PLC cells (Figure 2A). Western blot analyses revealed that Chx decreased the RCF + Ixz-induced increases in cl-PARP, poly-Ub, XBP1s, and LC3B levels but had little effect on CypA or B levels (Figure 2B). An important adaptive pro-survival UPR mechanism for cells to better recover from ER and proteotoxic stress is to shut down further protein synthesis via the PERK/eIF2α pathway (ER stress activates PERK to increase P-eIF2α levels and block protein synthesis) [10,11]. Western blot analyses revealed that RCF + Ixz decreased total (T)-PERK and P-eIF2α levels, suggesting that the pro-survival mechanism to shut down protein synthesis was blocked (Figure 2C). These results suggested that a possible mechanism for the RCF + Ixz-mediated increase in the death of HCC cells is by maintaining protein synthesis and further enhancing proteotoxic stress. Unlike in HCC cells, RCF + Ixz appeared to have little effect on the PERK/eIF2α pathway in EA non-cancer cells (Figure 2D). These results suggested RCF + Ixz may be less toxic in non-cancer cells due to little or no effects on apoptosis, autophagy, XBP1s, and PERK/eIF2α.

### 2.4. Inhibition of Autophagy Blocks the RCF + Ixz-Induced Increase in Apoptotic Cell Death in HCC Cells

SAR405, a selective Vps34 inhibitor that blocks autophagy [34], decreased the RCF + Ixz-induced death of Hep3B and HepG2 cells (Figure 2E and Appendix A). Western blot analyses revealed that SAR decreased the RCF + Ixz-induced increases in LC3B, cl-PARP, and XBP1s levels (24 h), whereas there was an increase in the levels of p62 (suggesting the inhibition of autophagy) and little or no effects on poly-Ub, CypA, or CypB levels (Figure 2F and Appendix A). These results suggested that effects of RCF + Ixz on autophagy (increase in Hep3B cells or decrease in HepG2 cells) were important in increasing apoptotic cell death.

### 2.5. Inducible Knockdown of the RCF Targets CypA and CypB Supports Pro-Survival Roles in Ixz- and RCF + Ixz-Treated HCC Cells

To determine the roles of CypA, CypB, and XBP1s in mediating the RCF + Ixz-induced death of Hep3B and PLC cells, we developed a doxycycline (Dox)-inducible lentivirus shRNA knockdown system (Figure 3A and Appendix A). The results indicated that the inducible knockdown of CypA increased the Ixz- and RCF + Ixz-induced death of Hep3B and PLC cells (Figure 3B and Appendix A), suggesting a pro-survival role for CypA. Inducible knockdown of CypB increased Ixz-induced cell death but not RCF + Ixz-induced cell death (Figure 3C and Appendix A). We suggest that since CypB levels were already decreased by RCF + Ixz, further knockdown would not have an effect. However, knockdown of CypB (mimics the RCF effect) and Ixz treatment did result in increased cell death, suggesting a pro-survival role for CypB.

### 2.6. Inducible Knockdown of XBP1s Supports a Pro-Survival Role Early but a Pro-Death Role Later

Although elevated levels of XBP1s can initially act as a pro-survival UPR factor, there is data suggesting that sustained XBP1s expression switches it to a pro-death factor [35,36,37,38,39,40,41,42]. The addition of Dox to induce the knockdown of XBP1s at time 0 h of RCF + Ixz treatment resulted in increased death of Hep3B and PLC cells. However, the addition of Dox 24 h after RCF + Ixz resulted in decreased cell death (Figure 3D and Appendix A). In all cases, adding Dox to negative control Hep3B/shScr or PLC/shScr cells did not result in changes in Ixz- and RCF + Ixz-induced cell death. The addition of IRE1α inhibitors to block the synthesis of its downstream modulator XBP1s decreased RCF + Ixz-induced cell death (Appendix A). These results suggest that XBP1s initially acts as a pro-survival factor but the maintenance of XBP1s at later times of RCF + Ixz treatment appears to act as a pro-death factor.

### 2.7. Inducible Expression of CypB Decreases and Inducible Expression of XBP1s Increases Cell Death in RCF + Ixz-Treated HCC Cells

To further determine roles for CypB and XBP1s in mediating the RCF + Ixz-induced death of Hep3B and PLC cells, we developed a Dox-inducible lentivirus expression system (Figure 4A,C). The results indicated that the inducible expression of CypB decreased RCF + Ixz-induced cell death in Hep3B and PLC cells (Figure 4B), suggesting a pro-survival role. Inducible expression (small increase) of XBP1s was only observed when Hep3B and PLC cells were treated with RCF + Ixz to create stress; in control-treated cells (no stress), there was no Dox induction of XBP1s expression (Figure 4C). The results revealed that the inducible expression of XBP1s increased RCF + Ixz-induced cell death in Hep3B and PLC cells (Figure 4D). An internal lentivirus plasmid protein TurboRFP was also induced by Dox in Hep3B/CypB and Hep3B/XBP1s cells, supporting the functionality of the system (Appendix A). In all cases, the addition of Dox to negative control (empty vector, EV) HCC cells did not result in changes in RCF + Ixz-induced cell death nor increased the expression of CypB, XBP1s, or TurboRFP.

### 2.8. An Orally Bioavailable RCF + Ixz Combination Reduces Hep3B Xenograft HCC Tumors In Vivo

To test the in vivo efficacy of the RCF + Ixz combination, we utilized the Hep3B xenograft HCC tumor mouse model system. The results indicated that the RCF (80 mg/kg body weight, daily) + Ixz (5 mg/kg, 2 days on/1 day off × 6) group had significantly reduced tumor volumes and final weights to a greater extent than in the RCF-, Ixz-, and vehicle control-treated mice (Figure 5A,B). Body weights between the four different groups did not vary significantly, suggesting a lack of general toxicity of RCF + Ixz (Figure 5C). The use of Ixz at 10 mg/kg was toxic. Western blot analyses of the final Hep3B tumors from the four groups provided limited information as to why the RCF + Ixz-treated tumors were smaller (Appendix A). Only CypB expression appeared lower in the RCF- and RCF + Ixz-treated tumors, suggesting that this may be a good drug response biomarker. There were no consistent differences in cl-PARP, poly-Ub, XBP1s, or CypA levels. We then separated RCF + Ixz-treated tumors into responders (*n* = 9) (average final tumor volume of 329 mm^3^ [range 132–669 mm^3^] and final tumor weight of 0.43 g [range 0.28–0.74 g]) and non-responders (n = 4) (average tumor volume of 903 mm^3^ [range 557–1368 mm^3^] and final tumor weight of 1.73 g [range 1.13–2.44 g]). Western blot analyses suggested increased cl-PARP, poly-Ub, and possibly XBP1s levels in responder tumors compared to non-responder tumors; no clear differences in CypA or B levels were noted (Figure 5D). The histological analysis revealed subtle differences between the four groups, with a possible increase in vacuolated (proteotoxic stress) cancer cells adjacent to blood vessels in RCF + Ixz-treated tumors (Appendix A). Overall, these results suggested that RCF + Ixz had anti-HCC efficacy in vivo without general toxicity and specific molecular differences (cl-PARP, poly-Ub, and XBP1s) were observed in vivo.

## 3. Discussion

Due to the increases in hepatitis C, obesity, and fatty liver disease, HCC has developed into a fast growing and lethal cancer with limited treatment options. Although there has been progress in the use of multiple tyrosine kinase inhibitors, immunotherapies, and novel combinations, the prognosis for many patients remains poor [43,44,45]. Our data identified a novel strategy for the treatment of experimental HCC models using RCF + Ixz to enhance ER and proteotoxic stress past a threshold pro-survival pathways cannot tolerate and therefore promote apoptotic cell death. We suggest that HCC (and other cancers) are vulnerable to excessive ER and proteotoxic stress induced by RCF + Ixz compared to normal cells/tissues, thus addressing a key limitation in the discovery of new agents and combinations, i.e., less toxic side effects.

Targeted therapies for solid cancers frequently fail due to inherent or acquired resistance [46]. Further improving the efficacy of targeted drugs is an important but challenging area of research for increasing the survival of patients with HCC and other solid cancers. A meta-analysis reveals that only 13.6% of total patients are eligible for targeted drugs and 7% benefit from targeted FDA-approved drugs [46]. It is likely that developing new, improved, and expensive targeted drugs will have limitations because few overall patients will benefit, and the problem of resistance reduces its application. The development and combination of drugs such as RCF with multiple targets (Cyp family) and Ixz (targets the UPS cellular pathway), in which toxicity is minimal and acquired resistance is likely more difficult to obtain, may be a more productive approach. Whether RCF + Ixz can result in acquired resistance needs further investigation. However, we have yet to find a cancer cell line (a total of 14, including prostate cancer, melanoma, and lung cancer lines [manuscript submitted]) that is inherently resistant to RCF + Ixz, suggesting that this strategy may increase patient eligibility and responses. Single cell sequencing of human multiple myeloma resistant to proteasome inhibitors identified CypA as a resistance gene, and the addition of CsA + proteasome inhibitor overcomes this resistance [30]. Therefore, further investigation of RCF (or other Cyp inhibitors) + Ixz (or other proteasome inhibitors) may be helpful in discovering better chemotherapeutic strategies against HCC.

A major limitation with current proteotoxic stress strategies (heat shock protein 90 or pan-histone deacetylase + UPS inhibitors) is toxicity to non-cancer cells and tissues [19,20]. RCF + Ixz maintains the HCC pro-death capability of this proteotoxic stress combination but without toxic effects on non-cancer cells and without general toxicity in vivo. We suggest that since multiple Cyp knockout mice are viable (CypA, B, or D), whereas heat shock protein 90 or multiple histone deacetylase knockout mice are not viable [47,48,49,50,51], the inhibition of Cyps in non-cancer cells will be expected to have limited or no toxicity. Because toxicity to non-cancer cells is more likely to result from Ixz, this allowed us to increase the RCF dose and decrease the Ixz dose without limiting anti-HCC efficacy. Therefore, we propose RCF + Ixz may be a new and promising proteotoxic stress strategy to improve responses in HCC but with limited toxicity.

There is strong evidence for using Cyp inhibitors against CypA and B isoforms in cancer chemotherapy because of their roles in tumor progression and stress protection [28,29]. CypD, another member of the Cyp family, is localized to the inner mitochondrial membrane and has an important role in mediating necrotic cell death by the mitochondrial permeability transition [48]. The concern that using Cyp inhibitors against CypD for cancer therapy will block necrosis is mitigated by evidence that inhibiting CypD and necrosis enhances apoptosis [52,53,54,55]. Significantly, the development of HCC is blocked in CypD knockout mice [56]. Moreover, the use of RCF and other Cyp inhibitors can inhibit the development of HCC in mouse models [21,22,57,58]. Therefore, the use of RCF and other Cyp inhibitors hold promise as single agents and in combinations with other drugs (e.g., RCF + Ixz) for the treatment of HCC and other cancers.

The UPR is an adaptive pro-survival mechanism responding to ER stress that activates kinase signaling pathways (IRE1α, PERK, and ATF6) to stop further protein synthesis (increases P-eIF2α levels) and enhance the folding capacity (increase the levels of the XBP1s/ CHOP/ATF4 positive transcription factors for chaperones) [10,11]. RCF + Ixz (and RCF alone) had immediate early effects on the expression of IRE1α and its downstream modulator XBP1s. At later times, when apoptosis begins to occur, the level of XBP1s was maintained at a higher level in RCF + Ixz-treated compared to RCF alone-treated HCC cells. Our data suggested that the maintenance of XBP1s expression by RCF + Ixz was a factor in the induction of cell death in HCC cells. Although elevated XBP1s expression can initially act as a pro-survival factor [10,11], there is data suggesting that sustained XBP1s expression switches it to a pro-death factor [35,36,37,38,39,40,41,42]. In non-cancer cells, XBP1s expression was not induced by RCF + Ixz, and there was minimal cell death. Presumably, the mechanism for keeping XBP1s levels low in stressed non-cancer cells must be stringently regulated in order to protect them from cell death. The idea of XBP1s maintenance as a pro-death mediator in cancer cells and ER stress biology is a much less well known and under-investigated topic of a drug therapeutic response and further investigations are required.

RCF + Ixz reduced the expression of PERK (major kinase for eIF2α) and blocked P-eIF2α, presumably allowing protein synthesis to continue under increased proteotoxic stress. The inhibition of protein synthesis blocked RCF + Ixz-induced cell death in HCC cells, supporting an important role for maintaining protein synthesis to increase efficacy. Interestingly, RCF + Ixz did not decrease PERK or P-eIF2α levels in EA non-cancer cells, suggesting this may be an important cancer-specific mechanism. We are not aware of another drug or combination with dual roles in maintaining XBP1s expression and decreasing PERK/P-eIF2α levels to result in cancer-specific cell death. In addition, the effect of RCF + Ixz on autophagy (whether to increase or decrease) appeared to play an important role in mediating its anti-HCC effects. Further investigations are required to decipher the importance of these multiple pathways in mediating the cancer-specific effects of RCF + Ixz.

The limitations of this study are the lack of (1) rigorous investigations of the potential toxicity of RCF + Ixz in the Hep3B xenograft model; (2) testing the RCF + Ixz combination in a more clinically relevant model of HCC; and (3) identification of the specific mechanisms of how (a) XBP1s increases apoptotic cell death in HCC cells but not in non-cancer cells at later times in vitro and in the Hep3B xenograft model and (b) the decrease in PERK levels prevents P-eIF2α to maintain protein synthesis during proteotoxic stress. Future studies will (1) test the RCF + Ixz combination in patient-derived HCC xenografts and organoid models; (2) use RNA sequencing and proteomic methods to identify potential XBP1s targets important in mediating apoptotic cell death in HCC experimental models; (3) develop inducible shRNA knockdown/protein expression or CRISPR-mediated deletion of PERK in HCC to determine if there is an important role in mediating RCF + Ixz-induced cell death; (4) determine in non-cancer cells whether increased XBP1s expression results in cell death; and (5) determine if RCF + Ixz can result in acquired resistance in HCC.

## 4. Materials and Methods

### 4.1. Reagents

RCF was obtained from Hepion Pharmaceuticals (Edmonton, AB, Canada); Ixz (MLN2238 biologically active [#A4008], the MLN9708 prodrug [#A4007]), and QVD (#A1901) were obtained from APExBIO (Houston, TX, USA); cycloheximide (#239765), SAR405 (#533063), doxycycline (#D9891), polybrene (#TR-1003-G), 2-hydroxypropyl-b-cyclodextrin (#H5784) were obtained from Sigma-Aldrich (St. Louis, MO, USA); cyclosporin A (#BML-A195) was obtained from Enzo Life Sciences (Farmingdale, NY, USA); thapsigargin (#sc-24017) was obtained from Santa Cruz Biotechnology (Santa Cruz, CA, USA); STF-083010 (#17370) and MKC-3946 (#19152) were obtained from Cayman Chemicals (Ann Arbor, MI, USA); and Coomassie blue (#20278), trypan blue (0.4%; #15250061), and puromycin (#J67236.8EQ) were obtained from Thermo Fisher Scientific (Waltham, MA, USA).

### 4.2. Cell Culture

The human HCC cell lines Hep3B (#HB-8064), PLC/PRF/5 (#CRL-8024), and HepG2 (#HB-8065), and non-cancer cells EA.hy926 (EA, umbilical vein; #CRL-2922) and dermal fibroblasts (HDFs; #PCS-201-012) were obtained from the American Type Culture Collection (ATCC, Manassas, VA, USA) and used within 6 months of resuscitation of the original cultures. HCC and EA cells were maintained in DMEM or EMEM (Thermo Fisher Scientific, ATCC) with 10% fetal bovine serum (R&D Systems, Minneapolis, MN, USA). HDF cells were maintained in Fibroblast Basal Medium plus Fibroblast Growth Kit-Serum Free (ATCC). All cell culture media contained 100 U/mL penicillin, 100 μg/mL streptomycin, and 0.25 μg/mL amphotericin (Thermo Fisher Scientific).

### 4.3. Drug Treatments

Cells were cultured in media containing RCF (2–10 μM), Ixz (2238, 9708; 10–100 nM), QVD (10 μM), cycloheximide (1, 10 μM), SAR405 (1, 2.5 μM), thapsigargin (10 nM), STF-083010 (50 μM), MKC-3946 (20 μM), or the DMSO (0.1%) control for varying times (10 min–72 h). The treatment of EA cells started when cells were confluent [59]. In all the experiments, floating and trypsinized attached cells were pooled for further analysis.

### 4.4. Trypan Blue Exclusion Assay to Measure Total Cell Death

Treated and control cells were harvested, resuspended in PBS, diluted 1:1 in 0.4% trypan blue, dead blue and live non-blue cells were immediately counted using a hemacytometer, and the % dead blue cells was determined from at least 2–3 independent experiments performed in triplicate.

### 4.5. Cell Proliferation Assay and Determination of the Synergy Combination Index

The CellTiter Aqueous colorimetric method from Promega (Madison, WI, USA) was used to determine the proliferation of HCC cells in media containing RCF (1–10 µM), Ixz (10–100 nM), cyclosporin A (1–10 µM), or the control (0.1% DMSO). Cell proliferation was normalized against the DMSO control and the data are expressed as percentages of the control from three independent experiments performed in triplicate. Whether drug interactions were synergistic, additive, or antagonistic was determined using the CalcuSyn Version 2 program from Biosoft (Cambridge, UK). This program is no longer available from Biosoft. A combination index (CI) ≤ 0.7 was synergistic.

### 4.6. Western Blot Analysis

The preparation of total protein lysates and western blot analysis were performed as previously described [60]. The following antibodies were used: cl-PARP (9541), XBP1s (D2C1F), LC3B (2775), eIF2α (9722), P-eIF2α (Ser51), PERK (C33E1), and IRE1α (14C10) from Cell Signaling Technology (Danvers, MA, USA); Ub (P4D1), p62 (D3), GRP78 (A10), actin (C-11), mouse anti-rabbit IgG-HRP (2357), and m-IgG-Fc BP-HRP (525409) from Santa Cruz Biotechnology; CypB (16045) and P-PERK (T982) from Abcam (Cambridge, MA, USA); CypA (SA296) from Enzo Life Sciences (Farmingdale, NY, USA); P-IRE1α (Ser724) from Novus Biologicals (Centennial, CO, USA); and TurboRFP (448252) from MyBiosource (San Diego, CA, USA). Precision Plus Protein Dual Color Standards (Bio-Rad Laboratories, Hercules, CA, USA) were used to estimate molecular weights in kDa. Markers were used to cut blots in horizontal strips so that high-, medium-, and low-molecular-weight targets could be analyzed separately with the appropriate antibodies. In some cases, after analysis, the strips were pretreated with methanol for 1 min, washed, treated with Ponceau S Staining Solution (Thermo Fisher Scientific) for 15 min to strip the antibody signal, and analyzed with a different antibody. After immunodetection, our preference for loading controls was staining of total proteins transferred to the membrane with Coomassie blue because drug treatments often affect the levels of typical housekeeping proteins such as actin or tubulin. Blot images were cropped for clarity of the presentation. Full uncropped blots are available in Appendix A.

### 4.7. Quantitative Real-Time Polymerase Chain Reaction (qPCR)

The procedure for qPCR was performed as previously described [61]. The following human DNA oligonucleotides from Thermo Fisher Scientific were used for qPCR: CypB sense 5′-AAGTCACCGTCAAGGTGTATTTT-3′ and antisense 5′-TGCTGTTTTTGTAGCCAAATCCT-3′ (153 bp amplicon) and XBP1s sense 5′-CCCTCCAGAACATCTCCCCAT-3′ and antisense 5′-ACATGACTGGGTCCAAGTTGT-3′ (101 bp amplicon) (Primer Bank [62]). Values from two independent experiments performed in duplicate were normalized to the housekeeping reference genes ribosomal protein large P0 (RPL0) sense 5′-GCAATGTTGCCAGTGTCTG-3′ and antisense 5′-GCCTTGACCTTTTCAGCAA-3′ (141 bp amplicon) and β-2-microglobulin sense 5′-ATGAGTATGCCTGCCGTGTGA-3′ and antisense 5’-GGCATCTTCAAACCTCCATG-3′ (101 bp amplicon) (Primer Bank [62]).

### 4.8. Inducible Knockdown of CypA, CypB, and XBP1s with Lentivirus Transduction

The shRNA design, lentivirus production, and infection were performed as previously described [63]. The following human DNA oligonucleotides (Thermo Fisher Scientific) targeting CypA, CypB, and XBP1s were cloned into the TET-pLKO puro DNA lentivirus vector (Addgene, Watertown, MA, USA; 21915 [64]): shCypA1 and shCypA2 [65]; shCypB1, CCGGCCTACGAATTGGAGATGAAGACTCGAGTCTTCATCTCCAATTCGTAGGTTTTTG; shCypB2, CCGGGCCTTAGCTACAGGAGAGAAACTCGAGTTTCTCTCCTGTAGCTAAGGCTTTTTG; shXBP1s-2, CCGGAGATCGAAAGAAGGCTCGAATCTCGAGATTCGAGCCTTCTTTCGATCTTTTTTG; and shXBP1s-3, CCGGGACCCAGTCATGTTCTTCAAACTCGAGTTTGAAGAACATGACTGGGTCTTTTTG. Validated sequences were obtained from the Mission shRNA database (Sigma-Aldrich). The control shRNA was TET-pLKO puro scrambled (Scr) DNA (Addgene; 47541 [66]). Media for virally transduced Hep3B and PLC cells contained 10% fetal bovine serum (Tet tested; R&D Systems) and puromycin (2 μg/mL). To induce the expression of shRNAs, doxycycline (100 ng/mL final; fresh preparation every 2 weeks) was added to the media.

### 4.9. Inducible Expression of CypB and XBP1s with Lentivirus Transduction

The CypB and XBP1s mRNAs were PCR amplified (Q5 high fidelity PCR kit E0555S; New England Biolabs, Ipswich, MA, USA) and cloned into the pCW57-RFP-P2A-MCS DNA lentivirus vector (Addgene; 78933 [67]). The following human DNA oligonucleotides containing a PstI restriction site at the 5′ end and a BamH1 site at the 3′ end (underlined) from Thermo Fisher Scientific were used to amplify cDNAs from Hep3B mRNAs: CypB forward 5′-TAAGCACTGCAGATGCTGCGCCTCTCCGAACGC-3′ and reverse 5′-TGCTTAGGATCCCTACTCCTTGGCGATGGCAAA-3′ (660 bp amplicon); and XBP1s forward 5′-TAAGCACTGCAGATGGTGGTGGTGGCAGCCGCG-3′ and reverse 5′-TGCTTAGGATCCTTAGACACTAATCAGCTGGGG-3′ (1131 bp amplicon) [34]. PCR-amplified bands were gel purified (QIAquick Gel Extraction Kit 28704; Qiagen, Germantown, MD, USA), digested with PstI–BamH1, and cloned into PstI–BamH1-digested pCW57-RFP-P2A-MCS. Lentivirus production, infection into Hep3B and PLC HCC cells, and doxycycline induction were as described above [63].

### 4.10. Treatment of Hep3B Xenograft-Bearing Mice with Orally Bioavailable RCF + Ixz

Athymic Nude-Foxn1^nu^ male mice were obtained from Envigo (Indianapolis, IN, USA). Hep3B cells (2 × 10^6^ cells in a 0.1 mL volume of 50% Matrigel (Cultrex Basement Membrane Extract Type 3; R&D Systems; 3632-010-02)) were subcutaneously injected into the dorsal hind flank. When tumors reached 50–100 mm^3^ (Mitutoyo 500-196-30 digital calipers), mice were randomly assigned to 4 groups, and treated by oral gavage (Instech Laboratories, Plymouth Meeting, PA, USA; FTP-20-38) with (1) the vehicle control (5% solution of 2-hydroxypropyl-b-cyclodextrin [vehicle for Ixz] or self-microemulsifying drug delivery system [vehicle for RCF; improves RCF solubility and oral bioavailability in blood] [68]) (n = 12); (2) RCF (80 mg/kg body weight, daily, 17 days) (n = 12); (3) Ixz (2238) (5 mg/kg, 2 days on/1 day off × 6) (n = 9); and (4) RCF80 + Ixz5 (RCF first followed by Ixz after 30 min) (n = 13). Tumor volumes (length × width × depth × 0.5236 mm^3^) were measured daily with digital calipers, and body weights determined every 7 days. On day 18, the mice were euthanized, the final tumor volume/weight were determined, and a portion was frozen for western blot analysis or placed in formalin for histology. Tumor sections were stained with H&E and analyzed under a microscope.

### 4.11. Statistics

Statistically significant differences between drug-treated and control cells were determined by two-tailed Student’s *t*-test (unequal variance) from 2–3 independent experiments performed in duplicate or triplicate, with *p* < 0.05 considered significant. Similarly, differences in the final Hep3B tumor volumes/weights and body weights were determined by two-tailed Student’s *t*-test (unequal variance) of the 4 treatment groups (*n* = 9–13). We used the *t*-test to compare 2 samples directly. For example, RCF + Ixz compared to RCF alone or to Ixz alone. The experimental data are presented as means ± standard deviations.

## 5. Conclusions

Discovering new strategies for the treatment of HCC is an urgent topic. Our RCF + Ixz combination strategy may take advantage of the vulnerability of HCC (and other cancers) to increases in proteotoxic stress. In addition to identifying the RCF targets CypA and B as mediators of RCF + Ixz efficacy, we suggest that the dual effect on the UPR to maintain XBP1s expression and protein synthesis (PERK/P-eIF2α) in HCC but not in non-cancer cells/tissues provides an important cancer-specific effect (Figure 6). In addition, we confirmed that orally bioavailable RCF + Ixz significantly inhibited HCC tumors in vivo without general toxicity. Since both RCF and Ixz are clinical trial-stage compounds [24,69,70], the clinical translation of our findings can be readily investigated.

## Figures and Tables

**Figure 1 ijms-26-06699-f001:**
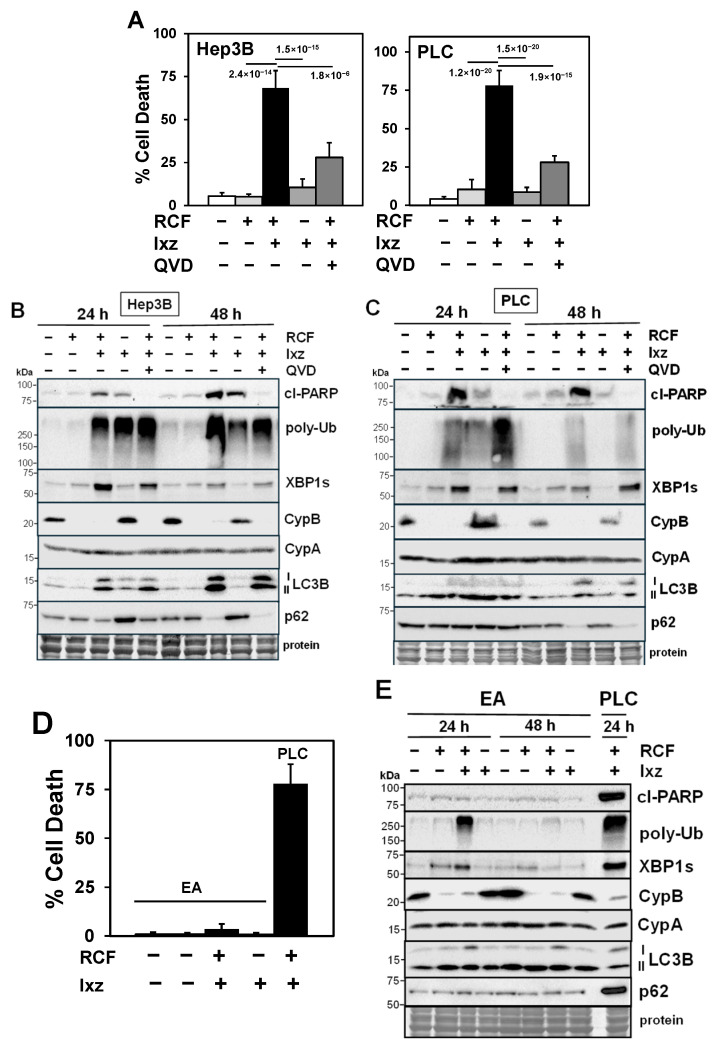
The RCF + Ixz combination enhances apoptotic cell death in HCC cells but not in non-cancer cells. (**A**) The trypan blue exclusion assay showed significantly higher cell death in RCF (5 μM) + Ixz (15 nM; 2238)-treated Hep3B and PLC cells (72 h) compared to RCF-, Ixz-, and control-treated cells. The addition of the apoptosis inhibitor QVD (10 μM) significantly decreased the RCF + Ixz-induced death of Hep3B and PLC cells. *p* values are shown near the bars. (**B**,**C**) Western blot analyses showed higher cl-PARP, XBP1s, and LC3B levels in RCF + Ixz-treated Hep3B (**B**) and PLC (**C**) cells (24 and 48 h) compared to RCF-, Ixz-, and control-treated cells. Poly-Ub levels were higher in cells treated with RCF + Ixz (48 h,) and CypB levels decreased with RCF and RCF + Ixz treatment (24 and 48 h). The decreased p62 levels observed with RCF + Ixz (48 h) suggest increased autophagy. No changes were noted in CypA levels. The addition of QVD to RCF + Ixz only decreased cl-PARP levels. (**D**) The trypan blue exclusion assay showed that RCF (5 μM) + Ixz (15 nM) slightly increased cell death in non-cancer EA cells (3.6%) compared to PLC HCC cells (78%). (**E**) Western blot analyses showed that RCF + Ixz only slightly increased cl-PARP, XBP1s, and LC3B levels in EA cells, unlike the greater increases in PLC HCC cells. In EA cells, there was an increase in poly-Ub levels (24 h) and decrease in CypB levels, but no clear differences in CypA levels. For all Western blots, the size of molecular weight markers in kDa are shown to the left. Protein refers to Coomassie blue staining after all immunological analyses were completed.

**Figure 2 ijms-26-06699-f002:**
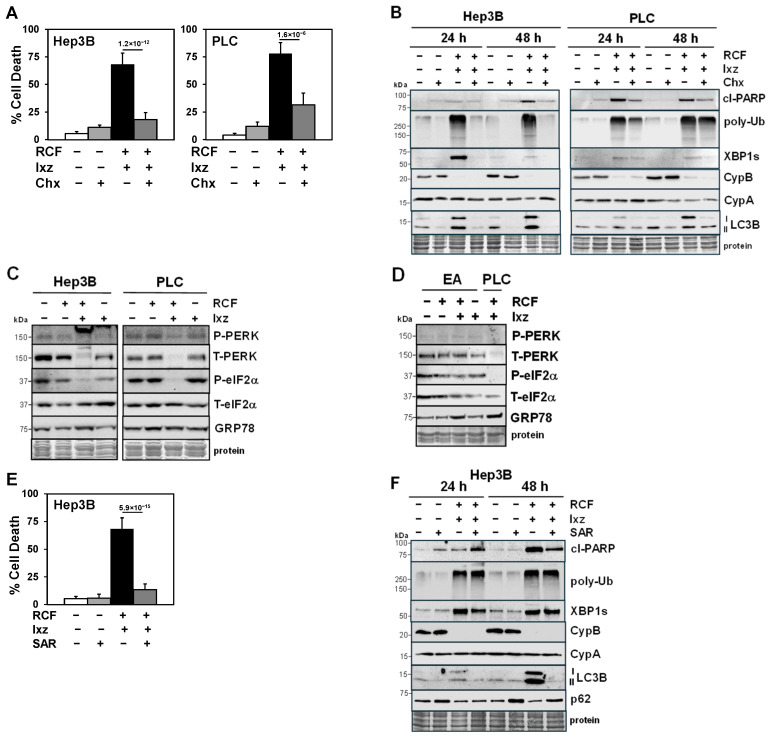
The protein synthesis inhibitor cycloheximide (Chx) and autophagy inhibitor SAR405 block the RCF + Ixz-induced death of HCC cells. (**A**) The trypan blue exclusion assay showed that the addition of Chx (10 μM Hep3B; 1 μM PLC) to RCF (5 μM) + Ixz (15 nM) significantly decreased the death (72 h) of Hep3B and PLC cells. *p* values are shown above the bars. (**B**) Western blot analyses showed that the addition of Chx decreased cl-PARP, poly-Ub, XBP1s, and LC3B levels in RCF + Ixz-treated Hep3B and PLC cells. No differences were noted in CypA or B levels. (**C**) Western blot analyses showed that RCF + Ixz decreased total (T)-PERK and P-eIF2α levels to greater extent than in RCF-, Ixz- or control-treated Hep3B and PLC cells, suggesting that the mechanism to decrease protein synthesis to better recover from proteotoxic stress was blocked. No clear differences in T-eIF2α or GRP78 levels were noted. (**D**) Unlike in HCC cells, RCF + Ixz did not decrease PERK or P-eIF2α levels in EA non-cancer cells. There was a small increase in GRP78 levels. (**E**) The trypan blue exclusion assay showed that the addition of SAR (2.5 μM) to RCF (5 μM) + Ixz (15 nM) significantly decreased the death of Hep3B cells (*p* value shown above the bar). (**F**) Western blot analyses showed that SAR decreased cl-PARP and LC3B levels and increased p62 levels (suggesting inhibition of autophagy) in RCF + Ixz-treated Hep3B cells (48 h). SAR decreased the RCF + Ixz-induced increase in XBP1s levels (24 h) but there were no clear differences in poly-Ub, CypA, or B levels. For all Western blots, the sizes of molecular weight markers in kDa are shown to the left. Protein refers to Coomassie blue staining after all immunological analyses were completed.

**Figure 3 ijms-26-06699-f003:**
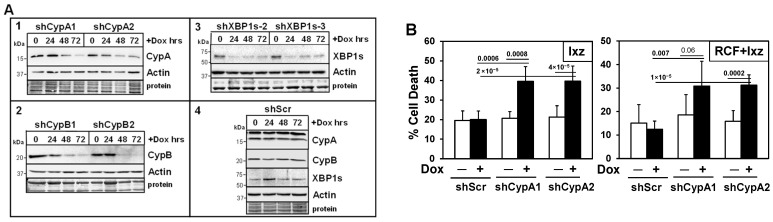
Inducible knockdown indicates that CypA and CypB are pro-survival proteins whereas XBP1s is a pro-survival protein early but pro-death protein later in RCF + Ixz-treated Hep3B HCC cells. (**A**) Western blot analyses of (**1**) Hep3B/shCypA-1, -2; (**2**) Hep3B/shCypB-1, -2; and (**3**) Hep3B/shXBP1s-2, -3 cells treated with Dox (100 ng/mL) for 24, 48, and 72 h. The results showed decreased CypA (**1**), CypB (**2**), and XBP1s (**3**) levels compared to control cells; no effect on actin levels was noted. Dox treatment of Hep3B/shScr negative control cells (**4**) showed no clear differences in CypA, CypB, XBP1s, or actin levels (same blot). Sizes of the molecular weight markers in kDa are shown to the left. Protein refers to Coomassie blue staining after all immunological analyses were completed. (**B**–**D**) Trypan blue exclusion assays showed that the addition of Dox (+) to induce knockdown of CypA (**B**) increased Ixz- (2238; 25 nM) (72 h) and RCF (5 μM) + Ixz (25 nM)-induced (27 h) cell death in Hep3B/shCypA-1 and -2 cells. The addition of Dox to induce the knockdown of CypB (**C**) increased cell death in Ixz- but not in RCF + Ixz-treated Hep3B/shCypB-1 and -2 cells. The addition of Dox to induce the knockdown of XBP1s (**D**) at time 0 h increased cell death in RCF + Ixz-treated Hep3B/shXBP1s-2 and -3 cells (27 h). However, the addition of Dox 24 h after RCF + Ixz decreased cell death after another 24 h. In all cases, the addition of Dox to Hep3B/shScr negative control cells did not result in differences in cell death. In addition, no Dox (−) cell death was similar to Scr negative control cells. In the shorter RCF + Ixz (27 h) treatments, cells were pretreated with Dox for 48 h. *p* values are shown above the bars.

**Figure 4 ijms-26-06699-f004:**
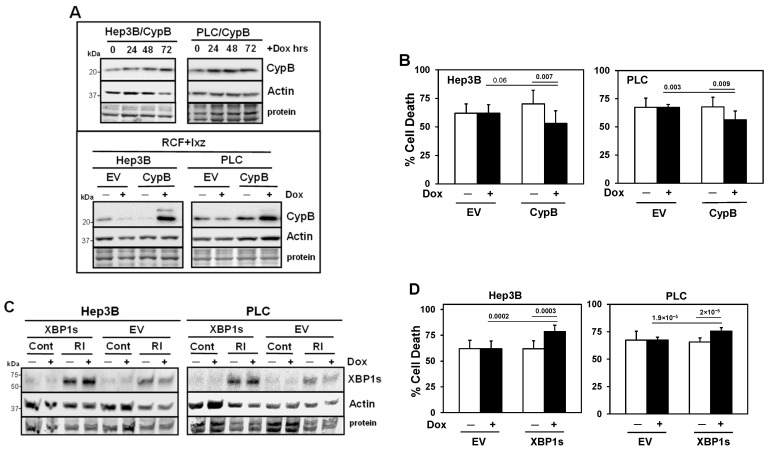
Inducible expression of CypB decreases and XBP1s increases cell death in RCF + Ixz-treated HCC cells. (**A**) Western blot analyses showed that the addition of Dox (24, 48, and 72 h) increased CypB levels but not actin levels in Hep3B/CypB and PLC/CypB cells compared to no Dox (0 h) (top panel). Bottom panel: the addition of Dox (+) increased CypB levels in RCF + Ixz (72 h)-treated Hep3B/CypB and PLC/CypB cells compared to empty vector (EV) negative controls and no Dox (−). No changes in actin levels were noted. (**B**) The trypan blue exclusion assay showed that the addition of Dox (+) to increase CypB levels decreased cell death in RCF (5 μM) + Ixz (15 nM) (72 h)-treated Hep3B/CypB and PLC/CypB cells compared to EV negative controls and no Dox (−). *p* values are shown above the bars. (**C**) Western blot analyses showed that the addition of Dox (+) slightly increased XBP1s levels only in RCF + Ixz (RI)-treated Hep3B/XBP1s and PLC/XBP1s cells compared to EV negative controls and actin. Control (Cont) DMSO-treated EV and XBP1s cells had very low levels of XBP1s −/+Dox. (**D**) The trypan blue exclusion assay showed that the addition of Dox (+) to induce XBP1s expression increased cell death in RCF + Ixz (72 h)-treated Hep3B/XBP1s and PLC/XBP1s cells compared to EV negative controls and no Dox (−). *p* values are shown above the bars. In (**A**) and (**C**), the sizes of molecular weight markers in kDa are shown to the left. Protein refers to Coomassie blue staining after all immunological analyses were completed.

**Figure 5 ijms-26-06699-f005:**
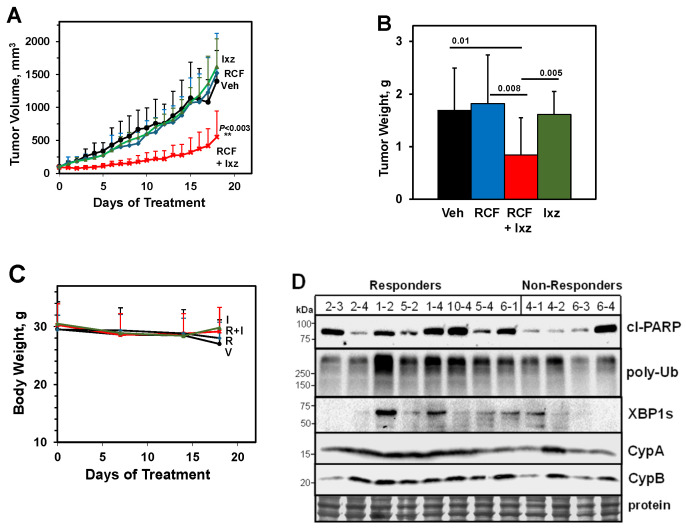
RCF + Ixz inhibit tumors in mice with Hep3B xenografts. (**A**) RCF (80 mg/kg, daily) + Ixz (5 mg/kg, 2 days on/1 day off × 6) significantly reduced the tumor volumes of Hep3B xenografts compared to RCF-, Ixz-, and vehicle (Veh) control-treated mice (**, *p* < 0.003). (**B**) Final tumor weights were significantly less after RCF + Ixz treatment compared to the RCF, Ixz, and Veh treatments (*p* values shown above bars). (**C**) No significant changes in body weights were observed in RCF + Ixz (RI)-, RCF (R)-, Ixz (I)-, and Veh (V)-treated mice. (**D**) Western blot analysis of Hep3B tumors in the RCF + Ixz group separated into responders (final tumor volume [mm^3^]; final tumor weight [g]) 2-3 [331; 0.43], 2-4 [403; 0.47], 1-2 [132; 0.28], 5-2 [669; 0.42], 1-4 [225; 0.37], 10-4 [221; 0.31], 5-4 [162; 0.39], 6-1 [486; 0.74] and non-responders 4-1 [912; 2.13], 4-2 [557; 1.13], 6-3 [774; 1.22], 6-4 [1368; 2.44]. Although there was variability between responders and non-responders, the results showed that responders had slightly increased cl-PARP, poly-Ub, and XBP1s levels compared to non-responders. No clear differences were noted in CypA/B levels.

**Figure 6 ijms-26-06699-f006:**
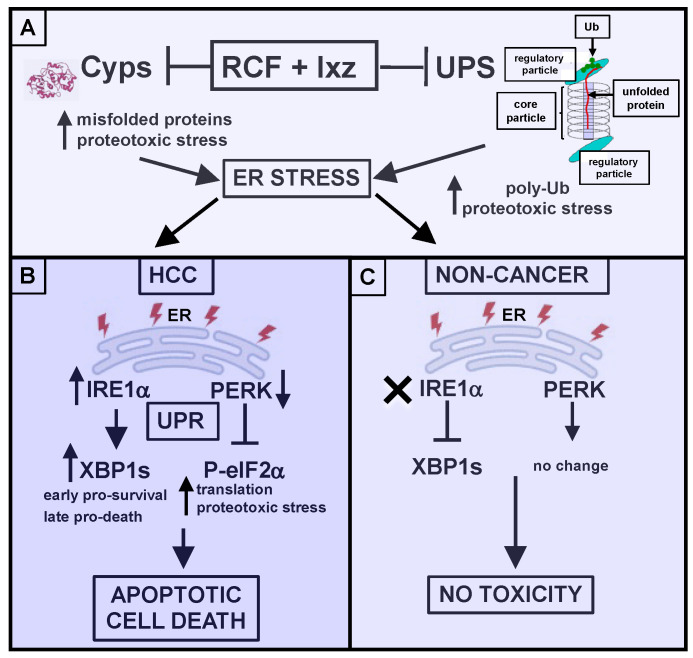
Schematic diagram of the pro-cell death effect of RCF + Ixz on HCC cells but not on non-cancer cells. (**A**) The combination of RCF (reduces CypB levels, inhibits CypA and other Cyp family members, resulting in increased levels of misfolded proteins and proteotoxic stress) + Ixz (blocks the UPS to increase poly-Ub levels and proteotoxic stress) amplifies ER stress. A reduction in CypB levels and increase in poly-Ub levels occur in cancer and non-cancer cells. (**B**) In HCC, RCF + Ixz activates the IRE1α/XBP1s UPR pathway, resulting in an early pro-survival effect. RCF + Ixz maintains XBP1s expression at later times, resulting in a pro-death effect. RCF + Ixz also decreases PERK levels, thus preventing P-eIFα from stopping further protein synthesis, resulting in a further enhancement of proteotoxic stress. The overall effect is to increase apoptotic cell death. (**C**) In non-cancer cells, even though RCF + Ixz increased poly-Ub levels and decreased CypB levels, there was no effect on IRE1α/XBP1s levels and the PERK UPR pathway, resulting in no cell death or toxicity.

## Data Availability

All data generated or analyzed during this study are included in this published article and Appendix A. Upon written or e-mail request, any resources or data will be made freely available.

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
