# Peer review of "A Novel Chemotherapy Combination to Enhance Proteotoxic Cell Death in Hepatocellular Carcinoma Experimental Models Without Killing Non-Cancer Cells"

_ijms, 2025, doi:10.3390/ijms26146699_

Round 1
Reviewer 1 Report
Comments and Suggestions for Authors
The manuscript by Perez-Stable et al., titled “A Novel Chemotherapy Combination to Enhance Proteotoxic Cell Death in Hepatocellular Carcinoma Without Killing Non-Cancer Cells,” presents a well-structured and scientifically novel investigation. The study demonstrates that combining rencofilstat, a pan-cyclophilin inhibitor, with the proteasome inhibitor ixazomib selectively enhances proteotoxic stress and induces apoptotic cell death in hepatocellular carcinoma cells, while sparing non-cancerous cells. This combination disrupts adaptive unfolded protein response pathways and overcomes pro-survival mechanisms, effectively reducing tumor growth in vivo with minimal systemic toxicity. I recommend that the authors make some revisions before the manuscript can be considered for publication in the International Journal of Molecular Sciences.
- The authors are advised to include the Hep3B xenograft model, or specify the experimental models used, in the title, as this is central to the novelty of the study.
- The term “proteosome inhibitor” is a misspelling and should be corrected to “proteasome inhibitor” for accuracyin the abstract.
- The authors should explain the roles of XBP1s and PERK in the abstract to improve clarity for readers unfamiliar with these pathways.
- The abstract does not specify the type(s) of non-cancer cells tested; including this information would helps to boost the readers interest.
- The authors should include the catalog numbers and suppliers for all proteins in the Reagentssection [5.1] to ensure clarity.
- The authors should include the catalog numbers for the cell lines in the Cell Culture section [5.2] for clarity and reproducibility.
- Did the authors measure tumor volume using calipers to assess length and width, as is commonly done?
- IIn Figuresoriginal and S1B and S2B, what concentration of poly-Ub did the authors use? It appears that a relatively high concentration may have been u
- The authors should briefly mention any study limitations or suggest future directions to demonstrate awareness of potential gaps.
- The authors should check for grammatical and typographical errors.
Author Response
- The authors are advised to include the Hep3B xenograft model, or specify the experimental models used, in the title, as this is central to the novelty of the study.
Thank you for the suggestion. We add “Experimental Models” into the title.
- The term “proteosome inhibitor” is a misspelling and should be corrected to “proteasome inhibitor” for accuracy in the abstract.
Thank you. We add the correct spelling.
- The authors should explain the roles of XBP1s and PERK in the abstract to improve clarity for readers unfamiliar with these pathways.
Thank you for the suggestion. In the revised abstract, we add the sentence “We then analyzed the effects of rencofilstat + ixazomib on XBP1s and PERK, critical factors in the unfolded protein response used by cells to survive proteotoxic stress.”
- The abstract does not specify the type(s) of non-cancer cells tested; including this information would helps to boost the readers interest.
Thank you for the suggestion. In the revised abstract, we add “umbilical vein or dermal fibroblast” non-cancer cells.
- The authors should include the catalog numbers and suppliers for all proteins in the Reagents section [5.1] to ensure clarity.
Thank you for the suggestion. The catalogue numbers for each item listed in section 5.1 have been added.
- The authors should include the catalog numbers for the cell lines in the Cell Culture section [5.2] for clarity and reproducibility.
Thank you for the suggestion. The catalogue numbers for Hep3B, PLC, HepG2, EA, and HDF listed in section 5.2 have been added.
- Did the authors measure tumor volume using calipers to assess length and width, as is commonly done?
Thank you for your observation. We add in section 5.10 “length × width × depth × 0.5236 mm3”.
- In Figures original and S1B and S2B, what concentration of poly-Ub did the authors use? It appears that a relatively high concentration may have been used.
Thank you for the inquiry. We used the anti-ubiquitin antibody at 1/10,000 dilution, as it results in a strong signal when proteotoxic stress is induced. This is a greater dilution than most antibodies used.
- The authors should briefly mention any study limitations or suggest future directions to demonstrate awareness of potential gaps.
Thank you for this suggestion. In the discussion, we add the paragraph “Limitations of this study are a lack of 1) rigorously investigating potential toxicity of RCF + Ixz in the Hep3B xenograft model; 2) testing the RCF + Ixz combination in a more clinically relevant model of HCC; and 3) identifying the specific mechanisms of how a) XBP1s increases apoptotic cell death in HCC but not in non-cancer cells at later times in vitro and in the Hep3B xenograft model; b) the PERK decrease prevents P-eIF2a to maintain protein synthesis during proteotoxic stress. Future studies will 1) test the RCF + Ixz combination in patient-derived HCC xenografts and organoid models; 2) use RNA sequencing and proteomic methods to identify potential XBP1s targets important in mediating apoptotic cell death in HCC experimental models; 3) develop inducible shRNA knockdown/protein expression or CRISPR-deletion of PERK in HCC to determine if there is an important role in mediating RCF + Ixz cell death; 4) determine in non-cancer cells whether increased XBP1s results in cell death; and 5) determine if RCF + Ixz can result in acquired resistance in HCC.
- The authors should check for grammatical and typographical errors.
Thank you for the suggestion. We have re-reviewed the manuscript for improvements in grammar and spelling errors and made the appropriate changes.
Reviewer 2 Report
Comments and Suggestions for Authors
Carlos and team did an interesting study on chemotherapy combination in HCC. I have following queries from the authors-
1 How the cut off of responders and non-responders were decided?
2 Authors claim lack of toxicity based on no change in body mass; toxicity can happen without body mass changes. Toxicological profiling is important to decide toxicity precisely. Are the number of days good enough to track body mass changes because of toxicity?
3 Since there seems no change in responder and Non responder of drug response. Actual mechanism of action might be different from what authors have proposed? What could be alternative explanation for responders and Non responders?
4 Please briefly explain the biological significance and role in cancer/ molecular biology of molecules shown in western blots (such as CypA, Cyp B, PARP, Poly-Ub) in introduction section, and relate it to your study and hypothesis. It will help reader understand more easily.
Comments on the Quality of English LanguageTitles are confusing and writing style doesn't seem like a connected story.
Author Response
Comment 1. How the cut off of responders and non-responders were decided?
Response 1. Thank you for the question. We arbitrarily identified RCF + Ixz responders (n=9) as having an average final tumor volume of 329 mm3 (range 132-669 mm3) and final tumor weight of 0.43 g (range 0.28-0.74 g). RCF + Ixz non-responders (n=4) had an average tumor volume of 903 mm3 (range 557-1368 mm3) and final tumor weight of 1.73 g (range 1.13-2.44 g). In the revised manuscript, we add this wording in section 2.7 to improve clarity.
Comment 2. Authors claim lack of toxicity based on no change in body mass; toxicity can happen without body mass changes. Toxicological profiling is important to decide toxicity precisely. Are the number of days good enough to track body mass changes because of toxicity?
Response 2. Thank you for your observation. We agree that no change in body weight is not a rigorous examination for lack of toxicity in the Hep3B xenograft model. In the discussion, we add limitations of the study to include a “lack of rigorously investigating potential toxicity of RCF + Ixz in the Hep3B xenograft model.” In addition, utilizing more clinically relevant models of HCC requiring longer treatments with RCF + Ixz may address whether body weight changes occur due to toxicity.
Comment 3. Since there seems no change in responder and Non responder of drug response. Actual mechanism of action might be different from what authors have proposed? What could be alternative explanation for responders and Non responders?
Response 3. Thank you for your observation and questions. We agree that a definitive mechanism in the Hep3B xenograft model was not addressed by our data. However, we believe there was a small trend towards increased cl-PARP (apoptosis), poly-Ub (proteotoxic stress), and XBP1s (pro-death UPR) in RCF + Ixz responders compared to non-responders by the Western blot data. Nonetheless, variability between samples makes it difficult to be confident of the specific mechanism. Alternative mechanisms would require immunohistochemistry analysis of proliferation (Ki67), apoptosis (cl-caspase-3), and proteotoxic stress (ubiquitin). In addition, RNAseq/proteomic analysis of tumor samples may provide further information of alternative mechanisms.
Comment 4. Please briefly explain the biological significance and role in cancer/ molecular biology of molecules shown in western blots (such as CypA, Cyp B, PARP, Poly-Ub) in introduction section, and relate it to your study and hypothesis. It will help reader understand more easily.
Response 4. Thank you for the suggestion. The introduction states “There is a clear rationale for using Cyp inhibitors against CypA and B isoforms in cancer chemotherapy because of their role in tumor progression and stress protection.” We add in line 88 “increased poly-Ub proteins” and in line 89 “increased cleaved-PARP” to help readers understand more easily.
Comments on the Quality of English Language: Titles are confusing and writing style doesn't seem like a connected story.
Thank you for the comments. We revised the main title to include “Experimental Models” as suggested by reviewer #1. In addition, we revised and simplified sub-titles in the Results sections 2.1-2.3; separated inducible CypA/B knockdowns (2.5) from inducible XBP1s knockdown (new 2.6); and added “Inducible Expression” in front of XBP1s (2.7) with the hope of reducing confusion and improving the connected story.